# TCFLE-8: a Corpus of Learner Written Productions for French as a Foreign Language and its Application to Automated Essay Scoring

**Rodrigo Wilkens[*], Alice Pintard[*], David Alfter[†], Vincent Folny[◇], Thomas François[*]**
[*]Cental, IL&C, UCLouvain,
[†]University of Gothenburg,
[◇]France Éducation international

## Abstract

Automated Essay Scoring (AES) aims to automatically assess the quality of essays. Automation enables large-scale assessment, improvements in consistency, reliability, and standardization. Those characteristics are of particular relevance in the context of language certification exams. However, a major bottleneck in the development of AES systems is the availability of corpora, which, unfortunately, are scarce, especially for languages other than English. In this paper, we aim to foster the development of AES for French by providing the TCFLE-8 corpus, a corpus of 6.5k essays collected in the context of the *Test de Connaissance du Français* (TCF - French Knowledge Test) certification exam. We report the strict quality procedure that led to the scoring of each essay by at least two raters according to the levels of the Common European Framework of Reference for Languages (CEFR) and to the creation of a balanced corpus. In addition, we describe how linguistic properties of the essays relate to the learners' proficiency in TCFLE-8. We also advance the state-of-the-art performance for the AES task in French by experimenting with two strong baselines (i.e., RoBERTa and feature-based). Finally, we discuss the challenges of AES using TCFLE-8.[1]

## 1 Introduction

Automated Essay Scoring (AES) aims to develop algorithms that can assess the quality of essays similarly to humans. The field may be traced back to the seminal work of Page (1966). Since then, several publications have been studying AES.[2] In the late 1990's, several functional AES systems were already available, either relying on Latent Semantic Analysis (e.g., Landauer et al. (1997)), NLP-extracted features combined with multiple regression (e.g., Burstein et al. (1998)) or Bayesian text classification (e.g., Rudner and Liang (2002)). As noted by Dikli (2006), a small amount of essays (less than 1000) could be enough for training such systems in some contexts. However, even collecting such a small corpus was difficult, as the essays need to be manually rated, and essays reliable assessment is a notoriously difficult task for humans (Wolfe et al., 2016).

Recent advances in AES have been made possible by Deep Learning (DL) approaches and large language models (Ramesh and Sanampudi, 2022). Prominent studies used embeddings (Alikaniotis et al., 2016), recurrent neural network (Taghipour and Ng, 2016), attention (Dong et al., 2017), and BERT-based architectures (Mayfield and Black, 2020). These approaches have also led to a growing need for large corpora.

Consequently, AES teams have turned their attention to learner corpus research, a branch of corpus linguistics providing large-scale, computerized, naturalistic learner production. Pioneering works such as the *International Corpus of Learner English* (ICLE) (Granger, 1993) and the *European Science Foundation L2 Database* (Perdue, 1993) demonstrated the potential of such learner data collections for Second Language Acquisition (SLA) research, but it is only recently that more learner corpora fitted for AES, i.e., large enough and annotated with different proficiency levels, were developed for various languages (Yannakoudakis et al., 2011; Blanchard et al., 2013; Geertzen et al., 2014; Wisniewski et al., 2013; Mendes et al., 2016; Sakoda and Hosoi, 2018).

Unfortunately, there is no such large corpus for French, making the situation for French AES far from encouraging. The first systems thus relied on unsupervised approaches: Lemaire and Dessus (2001) used Latent Semantic Analysis to compare

---

[1]TCFLE-8 is available at https://www.france-education-international.fr/corpus

[2]For comprehensive reviews see Ramesh and Sanampudi (2022); Lagakis and Demetriadis (2021); Klebanov and Madnani (2021); Uto (2021); Klebanov and Madnani (2020); Ke and Ng (2019); Shermis et al. (2013).

native language (L1) of student essays with textbook passages, whereas AUTO-EVAL (Zaghouani, 2002) automatically captured several L1 essay features, which are heuristically combined. More recently, Parslow (2015a) trained a Naive Bayes classifier on a very small corpus of 200 essays written in foreign language (FL). Finally, Ranković et al. (2020) were the first to fine-tune BERT for FL French AES on more data, but they did not release it and only a single L1 is represented.

Therefore, in order to support the development of AES solutions for French, the need for a large and reliable corpus of written French essays becomes apparent. In this paper, we make two main contributions. First, we provide the community with the TCFLE-8 corpus[3], composed of 6,569 learner essays, with 8 different languages of habitual use, scores, from at least 2 raters, for the 6 levels of the Common European Framework of Reference for Languages (CEFR) (Council of Europe, 2001), and automatically annotated with +5k features. These essays were collected in the context of the official French Knowledge Test (TCF) exam, one of the main certification exams for French. Second, we provide solid baselines for future research in AES using TCFLE-8. This is the largest AES modeling study that has been done for French.

This paper is organized as follows. Section 2 presents the main characteristics and uniqueness of corpora used for AES. We then present the corpus compilation process, presenting the official certification exam which our corpus is based on and the essay selection process (Section 3). Next, in Section 4, we present the TCFLE-8 corpus, discussing its size, metadata and annotation. An exploration of TCFLE-8 for AES systems is presented in Section 5. Finally, final remarks are presented in Section 6. TCFLE-8 is freely available for research purposes.

## 2 Related Work

Developing a French corpus for AES means taking part in the field of learner corpus research, which, since its emergence in the late 1980s, gave rise to more than 200 learner corpora around the world[4]. Reviews on learner corpora (Gilquin,

---

[3] *Test de connaissance du français* (French knowledge test); FLE stands for *français langue étrangère* (French as a foreign language) and 8 refers to the eight different languages of habitual use included in the corpus.

[4]See the *Learner Corpora around the World* (https://uclouvain.be/en/research-institutes/ilc/cecl/learner-corpora-around-the-world.html) for corpora that have been described in scientific publications.

2015; Granger et al., 2013) point out the prevalence of English as target language for more than half of the corpora, the rest focusing on German (Siemen et al., 2006; Gut, 2012; Belz, 2004), Spanish (Lozano, 2009; Cestero Mancera et al., 2002), French (Granger, 2003; Granfeldt et al., 2006), Italian (Di Nuovo et al., 2022) and others (Atwell and Alfaifi, 2014; Wang et al., 2015; Martin et al., 2012). In terms of usual or native language of the learners (L1), the majority of learner corpora are mono-L1. Multi-L1 corpora are however favored today because they allow to study the influence of various L1s on the target language and they offer a wider degree of generalization.

Among this large body of written learner corpora, we will focus on two types relevant for this work: corpora used for AES and learner corpora targeting French as a foreign language.

### 2.1 Corpora for AES

Corpora built for AES can focus of specific dimensions, such as the organizational skill of essay writing (e.g., ICLE (Granger, 1993)) or the persuasive nature of the essay (e.g., *Argument Annotated Essays* (Stab and Gurevych, 2014)), but they usually address the global level of a learner production on a proficiency scale. One of the most commonly used scales for foreign languages is the Common European Framework of Reference for Languages (CEFR) (Council of Europe, 2001), describing six levels of proficiency from A1 (beginner) to C2 (advanced). As the mapping between each essay and its proficiency level is critical in AES, it is best to use essays written in the context of official L2 certifications, as they benefit from strict rating procedures, usually with at least two professional raters grading each production. We distinguish these corpora, which we call *candidate corpora*, from other *learner corpora* containing productions collected in language classes or on web forums.

#### 2.1.1 Candidate corpora

In addition to having more reliable proficiency ratings, candidate corpora also contain more varied learner profiles in terms of L1, age and background. The largest candidate corpus is the *Cambridge Learner Corpus* (CLC) (Nicholls, 2003) with more than 50 million words from 200,000 written productions and 138 different L1s. It was compiled from English exams of Cambridge Assessment English and two subparts of this corpus are available for research. First, the *OpenCLC* (Lexical Com-

puting Limited, 2017) is composed of more than 10,000 texts from candidates of 7 different L1s. The second available subpart of the CLC, *First Certificate of English* (CLC-FCE), contains 1,238 texts aligned with the CEFR (Yannakoudakis et al., 2011; Vajjala and Rama, 2018). Another corpus targeting English was released by Educational Testing Service, the *ETS corpus of non-native written English* or TOEFL11 (Blanchard et al., 2013). Initially compiled to perform L1 detection tasks, this corpus was later used in AES to explore both traditional machine learning (Rupp et al., 2019) and deep learning models (Nadeem et al., 2019). It contains 12,100 English essays written by TOEFL candidates of 11 non-English native languages. The essays are presented with their prompt and proficiency level given by ETS (low-medium-high).

Collaborations between certified testing organisations and research groups also developed for other European languages, resulting into three recent candidate corpora. The *MERLIN* corpus (Wisniewski et al., 2013) contains 2,290 written productions from standardized tests targeting German, Italian (TELC institute) and Czech (UJOP Institute). Its design allowed for cross-lingual AES experiments (Vajjala, 2018; Arhiliuc et al., 2020; Bestgen, 2020; Caines and Buttery, 2020). The *COPLE2* corpus (Mendes et al., 2016), containing 966 essays written in Portuguese (ICLP and CAPLE institutes), and the *ASK* corpus (Tenfjord et al., 2006b), with 1,936 texts written by candidates to the Norwegian Language Test, have also both been used for AES (del Río et al., 2016; Berggren et al., 2019; Carlsen, 2012). Table 6 in Appendix A provides an additional detailed description of existing candidate corpora.

### 2.1.2 Corpora from language classes

Some corpora compiled from learner productions in language classes also prove to be suitable for AES tasks. For example, EFCAMDAT (*Education First-Cambridge Open Language Database*) (Geertzen et al., 2014) has been used in AES to investigate features related to the CEFR scale (Arnold et al., 2018), to classify based on errors (Ballier and Gaillat, 2016) or neural AES models (Kerz et al., 2021). To the best of our knowledge, this is the largest L2 corpus used in AES that does not come from certification exams. EFCAMDAT contains 83 million words from more than 1 million essays written by learners of Education First's online English school. These essays span 16 levels

traceable to the CEFR scale, and the prompts are level-specific (Geertzen et al., 2013).

AES experiments were also conducted for Spanish on *CEDEL2*, a learner corpus of more than 1 million words from 4,399 learners of 11 different L1s (Lozano, 2009), for Swedish on the *SweLL* corpus containing approximately 600 texts (Volodina et al., 2016) and for Japanese on the *I-JAS* corpus of texts written by 1000 learners of 12 different native languages (Sakoda and Hosoi, 2018). These experiments involve traditional machine learning work with features (del Río et al., 2016; Pilán and Volodina, 2018; Lee and Hasebe, 2020) or deep learning (Lilja, 2018; Ruan, 2020; Hirao et al., 2020a).

### 2.2 Learner corpora targeting French

To the best of our knowledge, there are no candidate corpora for French. Most learner corpora targeting French were compiled to study interlanguage[5] (Selinker, 1972). They were collected from language courses at university, so the levels represented are mainly intermediate and advanced. The *French Interlanguage Database* (Granger, 2003) contains 450,000 words. Other corpora designed for interlanguage investigation include the *Learner Corpus French* (Vanderbauwhede, 2012), containing 500,000 words, and the *Chy-FLE/Hellas-FLE* (Valetopoulos and Zając, 2012), containing 150,000 words. The *Corpus Interlangue* (Gaillat and Roa, 2020), a written/spoken and bilingual corpus, contains texts and interviews from 115 students. The *Corpus Ecrit de Français Langue Etrangère* (Granfeldt et al., 2006) approaches learners interlanguage in the language development sequences. It is the only corpus representative of all proficiency levels for French, and it contains 100,000 words. It has been used for AES to find the features most correlated with CEFR levels (Parslow, 2015a). Finally, the French part of the *Word Reference Corpus* (Berdicevskis, 2020) constitutes the largest learner corpus for French with 4 million words from forum posts on the Word Reference website. It has been used to study contact-induced simplification, but despite its considerable size, it was not used for AES, because it is noisy and text levels have not been evaluated. More information on learner corpora targeting French is presented in Table 6 in Appendix A.

---

[5]Interlanguage describes the unique linguistic organisation developed by a foreign language learner, which presents some features of previously acquired language and may overgeneralize L2 patterns.

## 3 Corpus compilation

### 3.1 Data collection

TCFLE-8 being a candidate corpus for French, it has been collected by one of the agencies carrying out official certification in L2 French: *France Education International* (FEI). FEI is a French agency under the supervision of the Ministry of National Education and Youth. With a workforce of over 250 employees and a network of more than 1,000 experts, FEI acts in various fields of cooperation in education and training and contributes to the promotion of the French language and the French-speaking world. FEI offers a wide range of certifications in French aligned with the six CEFR levels: initial diploma in French language (DILF), diploma in French language studies (DELF), diploma in advanced French language studies (DALF) and French knowledge test (TCF). Around 650,000 candidates take one of these examination on an annual base in more than 180 countries.

As its name implies, TCFLE-8 is based on the TCF, a linear test aligned with the six CEFR levels. The TCF is used mainly for academic studies, migration purposes and citizenship. Its written component, made up of three independent tasks, is taken annually by 120,000 candidates, 60% of which sit their exam on computer.

The correction is performed by professional raters. FEI has a pool of about 100 raters, recruited on occupational profiles (experienced teachers, previous experience for rating with French). Applicant raters take a psychometrically-calibrated rating competence test for writing and attend a two-day training. At the end of this procedure, the recruitment is confirmed or not. To ensure reliability in the long term, reliability indices of raters are assessed annually, and a decision is made regarding whether to retain them in the pool. In addition, to ensure reliability at the candidate level, FEI adopts a double rating approach.[6] In case of discrepancy, a third rater is called to independently rate the 3 productions. The final level of the candidate is established based on the frequency of the CEFR levels given to the three candidate's productions.

To identify the CEFR level, the raters use adapted CEFR descriptors and scales. The descriptors (and the rating) are holistic, although each descriptor is aiming at linguistic (organisational), pragmatic and sociolinguistic dimensions and their related criteria. Until now, language test providers used both analytical and holistic scales (Hamp-Lyons, 1995). There is no clear consensus on the superiority of one type of scale in terms of reliability and efficiency (Ono et al., 2019).

Language competence is multidimensional (Bachman, 1990; Bachman and Palmer, 2010; Oller and Hinofotis, 1980; Vollmer and Sang, 1983) and is a measurable skill (Vollmer and Caroll, 1983). Measuring writing skill implies considering various facets: candidate proficiency, rater leniency/harshness and difficulty of the task. To this aim, "Many-facet Rasch measurement (MFRM) is a psychometric approach that establishes a coherent framework for drawing reliable, valid, and fair inferences from rater-mediated assessments, thus answering the problem of fallible human ratings" (Eckes, 2009). Therefore, we applied MFRM to the FEI dataset of TCF exams in order to identify and avoid the fallible human ratings in the data set.

### 3.2 Data cleaning

The original data collected by FEI had to be cleaned at various levels. First, outlier identification consisted in removing candidates' responses that did not achieve the A1 level, were copies of the prompt, too short/long, or off-topic. Next, we leverage the Rasch information in the dataset to detect texts for which human raters might have failed to provide a reliable judgment. We compared FEI raters' original scores and the scores adjusted by the Rasch method, using standardized residuals. After an empirical evaluation, we removed all essays with a standardized residual value greater than 4.[7] In addition, we also dropped essays with a low confidence assessment (e.g., candidates that are on the borderline between levels). To accomplish this, we removed all cases where both raters disagree with each other and with the candidate's final score, and we also removed the cases where there is a distance of three CEFR levels between the lowest and the highest ratings. After this process, we set the essay score as the candidate's CEFR level when at least one of the raters assigned that level to the essay. Alternatively, if both raters agreed with the essay's score, we duly assigned this level to the essay. Any essay that does not fit any of these two criteria has been removed.

---

[6]It has to be mentioned that the rating of the set of the 3 tasks is done by the same rater, thus not being independent.

[7]In our empirical evaluation, we explored four standardized residue values (2, 3 and 4), observing that around 5.6% of the corpus has a standardized residue of 2, 0.9% has a value of 3 and 0.4% a value of 4.

After outlier removal, the next step was to get a representative sample from the set of TCF essays available. For a fair representation, the level of the text is an obvious variable to control. In addition, we controlled for the language of habitual use[8], aiming for a representation of the most frequent languages. As the top five were all European ones and the 6th was Kabyle, a Afro-Asiatic language, we also included Chinese and Japanese to get a better representation of various typological families of languages. Thus, we launched a random sampling controlling for the 6 CEFR levels and the candidate's language of habitual use. To apply this algorithm, we set up an objective function that approximates the CEFR scores distribution by language. In order to reflect the distribution in the whole dataset, we divided the current language into frequency bands: very frequent (English and Arabic), frequent (Spanish, Kabyle, Portuguese and Russian) and infrequent (Chinese and Japanese) languages. For a description of the resulting corpus see Section 4.

### 3.3 (Pseudo-)anonymization

Candidates sometimes include personal information in their essays. While this does not pose a problem for the assessment, it can expose candidates when the texts become public. This exposure is generally tackled with anonymization methods (e.g., Wisniewski et al. (2013); Mendes et al. (2016); Tenfjord et al. (2006a); Gablasova et al. (2019); Rakhilina et al. (2016)) or pseudo-anonymization (e.g., (Glaznieks et al., 2020; Preradovic et al., 2015; Rosen et al., 2020; Dirdal et al., 2022)) in the literature on learner corpora. Typically, these processes capture names (e.g., Gablasova et al. (2019); Preradovic et al. (2015); Rosen et al. (2020); Rakhilina et al. (2016)), but sometimes they also capture other information, such as location and date (e.g., Glaznieks et al. (2020); Tenfjord et al. (2006a); Wisniewski et al. (2013) ), geo-data (e.g., Volodina et al. (2019)) and language-specific substitutions (e.g., Wisniewski et al. (2013)). In our work, we decided to provide anonymous and pseudo-anonymous versions of TCFLE-8. The latter is intended to provide a more natural text, but pseudo-anonymization may introduce grammatical errors (e.g., wrong contractions).

This work uses the MAPA tool[9] (Gianola et al.,

2020) for (pseudo-)anonymization. With this tool, we target 7 entities: names, address (i.e., country, city, building, territory and place), date (i.e., day of week, month, year and day), e-mail, organization, amount and phone. After the pseudo-anonymization, we assessed its quality.[10] During this process, we noticed some consistent flaws in the tool that were corrected in the corpus to improve its quality. The observed issues consisted of an overanonymization of words at sentence beginnings when predicated by a name, and an omission to replace email addresses.

## 4 The TCFLE-8 corpus

At the end of the compilation process, the final TCFLE-8 corpus comprises 6,569 essays (581,333 words). Some figures about the corpus size by CEFR levels are shown in Tables 1, 2, 3 and 4. It is expected that beginner-level essays tend to be shorter than the other ones (Frase et al., 1998). Moreover, the extreme levels (A1 and C2) are less represented in the corpus. This might be caused by two factors: (1) few A1-level learners seek an official language exam since this level is rarely sufficient for official purposes (e.g., employment and visa requirements), and (2) reaching the C2 level in a foreign language is extremely difficult.

| | #essays | %essays | avg #wrd (stdev) |
|---|---|---|---|
| A1 | 689 | 10.49 | 69.78 (34.02) |
| A2 | 1375 | 20.93 | 91.69 (44.80) |
| B1 | 1466 | 22.32 | 119.11 (49.89) |
| B2 | 1427 | 21.72 | 133.61 (44.92) |
| C1 | 1127 | 17.16 | 133.92 (45.91) |
| C2 | 485 | 7.38 | 138.92 (48.45) |
| *Total* | 6569 | 100.0 | 119.67 (50.31) |

Table 1: Description of the TCFLE-8 corpus by CEFR level: number of essays, percentage, and mean and standard deviation of word number per essay.

Table 2 indicates the number of essays distinguishing the gender. It is interesting to note that about 58% of the sample is composed of women and this proportion is not the same at at each level. Table 3 shows the essays by three tasks in the TCF exam, where there is no general difference between one task and the others. Finally, Table 4 picture the amount of essays in the different languages of habitual use.

---

[8]The language of habitual use is the language the candidate indicates as the one they usually use.

[9]https://gitlab.com/MAPA-EU-Project/

[10]The evaluation scores are presented in Section B.

| (CEFR) Level | Men | Women |
|---|---|---|
| A1 | 394 | 295 |
| A2 | 574 | 801 |
| B1 | 596 | 870 |
| B2 | 537 | 890 |
| C1 | 441 | 686 |
| C2 | 198 | 287 |
| Total | 2740 | 3829 |

Table 2: Number of essays according to the gender, per (CEFR) level

| (CEFR) Level | Task 1 | Task 2 | Task 3 |
|---|---|---|---|
| A1 | 225 | 223 | 241 |
| A2 | 426 | 413 | 536 |
| B1 | 447 | 475 | 544 |
| B2 | 485 | 485 | 457 |
| C1 | 358 | 431 | 338 |
| C2 | 156 | 173 | 156 |
| Total | 2097 | 2200 | 2272 |

Table 3: Number of essays according to the task number

| Lang. | A1 | A2 | B1 | B2 | C1 | C2 |
|---|---|---|---|---|---|---|
| JPN | 8 | 135 | 171 | 170 | 48 | 2 |
| CHI | 34 | 165 | 244 | 189 | 45 | 4 |
| SPA | 124 | 187 | 175 | 182 | 178 | 58 |
| ARA | 135 | 160 | 163 | 153 | 160 | 135 |
| POR | 102 | 187 | 182 | 191 | 172 | 38 |
| ENG | 125 | 163 | 167 | 165 | 169 | 128 |
| RUS | 103 | 198 | 183 | 196 | 180 | 29 |
| KAB | 58 | 180 | 181 | 181 | 175 | 91 |

Table 4: Number of essays according to the language of habitual use (language code follows ISO639-2)

Comparing TCFLE-8 to existing corpora (see Table 6 in Appendix A), it is the largest French learner corpus suitable for AES – both in size and L1 representation –, the third largest candidate corpus to our knowledge and its annotation layers provide the richest information (see Section 4.2). It also covers all 6 CEFR levels.

## 4.1 Metadata

As a complement to the text of the essays and their CEFR scores, assigned according to the procedure described in Section 3.2, the TCFLE-8 corpus provides information about the candidate who wrote the essay and the essay prompt.

Regarding the candidates' information, their gender and language of habitual use are provided. The candidates communicate this information when they register for the TCF exam. Overall, the corpus contains slightly more women than men (58% vs 41%), see Table 2. As for the language of habitual use, the corpus covers 8 languages, as described in Section 3.2: English, Arabic, Spanish, Russian, Portuguese, Kabyle, Chinese, and Japanese, respectively, with 917, 906, 904, 889, 872, 866, 681, and 534 essays (see Table 4.

The CEFR level achieved by each candidate considering the three written productions is also reported. This is the official CEFR level assigned to the candidate for the written part of the TCF exam.

The Quadratic Weighted Kappa (QWK) between the candidate's level and the CEFR-level of the essay is 0.98. It is expected that this value should be high, but not equal to 1, due to the cases where candidates cannot maintain a consistent level of essay quality. In addition, the scores assigned by the FEI raters in the double rating procedure are also available. They have a QWK of 0.71 (correlation of 0.93) with each other and 0.84 (correlation of 0.85) with the CEFR-level of the essay.

Finally, the prompt and its position in the sequence of three TCF tasks are also reported. This information, which relates to the exam, contextualizes the essay's input and the grading sequence (described in Section 3.1). Regarding the prompt position, it is balanced in TFCFL-8 (32% essays for the first task, 33% for the second, and 35% for the third). Table 3 shows the distribution of the three tasks across the six CEFR levels.[11]

## 4.2 Essay annotation

In addition to the above metada, TCFLE-8 also includes a linguistic annotation layer aimed to describe the learners' proficiency. This annotation was automatically performed using the FABRA toolkit (Wilkens et al., 2022). It allows computing the distribution of over 400 linguistic variables grouped by family of related variables (e.g., lexical diversity, and lexical frequency). These distributions are aggregated using 18 statistical descriptors, which results in more than 5k annotations per essay. In addition, we extended the existing FABRA features by including others related to SLA. In par-

---

[11] We calculated the correlation – Spearman for continous variables or Point-biserial for binary variables – between the CEFR score and the metadata described above to identify possible biases in the scores; all correlations were between 0.038 and 0.09. This analysis confirms that the sampling process did not induce unexpected biases.

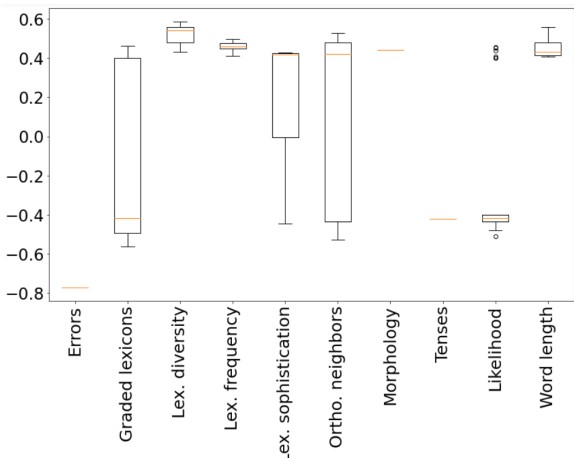

Figure 1: Box-plot of feature correlations by family.

ticular, we included the error annotation provided by *Language tool*[12], which includes, among others, the identification of agreement, casing, grammar, typography, punctuation, and typos. We also included pedagogical annotation based on the work of Pintard and François (2020) for extending the CEFR level-related vocabulary. It should be noted that, as the pseudo-anonimzation process may alter text properties, we have chosen to perform this feature extraction on the original essays.

In order to better characterize how linguistic properties present in TCFLE-8 are associated with the learners' proficiency, we computed Spearman's correlations between each of above feature (gathered by the families in Wilkens et al. (2022)) and the essays' CEFR level. In the process, we dropped features with correlations lower than 0.4 or p-values higher than 0.05. Next, as many features are variants of each other, we calculated the correlation matrix within each family to identify redundant features (i.e., an absolute correlation above 0.90). Finally, for each set of similar features, we considered only the one most correlated with the CEFR level. After this procedure, we kept 119 correlated features. In Figure 1, we show their distribution by family of variable.[13]

Our analysis of the selected features highlighted linguistic properties of essays already reported in studies investigating foreign language writing. For example, measures of word length have been known to be good predictors of proficiency level for English (Ferris, 1994; Grant and Ginther, 2000), for

---

[12]https://pypi.org/project/language-tool-python/
[13]Table 9 in the appendices presents the list of all correlated features and their correlation values.

Swedish (Pilán and Volodina, 2018), for Japanese (Hirao et al., 2020b) and for French Parslow (2015b). As regards lexicon, diversity measures (e.g. type/token ration) correlate with proficiency in English (Lu, 2012; Vajjala, 2018), whereas sophistication measures based on word frequencies have yielded similar results in a number of studies: more proficient writers use, on average, fewer frequent words (Laufer and Nation, 1995; Attali and Burstein, 2006; Crossley and McNamara, 2012; Guo et al., 2013). Finally, the most discriminating feature in TCFLE-8, namely the error-rate, is also one of the most correlated features to proficiency levels in the CLC-FCE and TOEFL11 (Yannakoudakis et al., 2011; Vajjala, 2018). In addition, while this analysis confirms existing research findings in AES, it also points out that TCFLE-8 may be helpful for new SLA studies. Indeed, we also provided several features explored in other acquisition-related fields, such as the OLD20 (measuring orthographic similarity) (Coltheart et al., 1977; Yarkoni et al., 2008), which are significant in our corpus but had not been linked to L2 writing proficiency so far, to the best of our knowledge.

## 5 AES for French

In this section, we analyze the applicability of the TCFLE-8 corpus for training AES systems. For this purpose, we explore two approaches: deep learning, since most AES systems relied on neural networks (Ramesh and Sanampudi, 2022), and feature-based machine leaning.

We split the anonymized corpus with 80% for training, 10% for validation, and 10% for testing, stratifying by score and language. In addition, to explore the impact of model initialization, we performed 5 repetitions of the training process; in each one, we adjusted the test set so that it does not overlap with the others. We performed a hyperparameter exploration using the accuracy on the validation set.

For the deep learning model, we used CamemBERT (Martin et al., 2020), a RoBERTa-based model for French. As for the hyperparameters[14], we use a learning rate of 5e-5 and an early stop of 5. For machine learning, we use the XGBoost[15] and

---

[14]The hyperparameters search explored 1e-4, 5e-5, 1e-5, 5e-6 and 1e-6 as learning rate, 1, 3, 5, 7 and 10 as early stop patience, searching up to 40 epochs.
[15]The hyperparameters used for XGBoost and the values explored are gbtree as booster, alternatively exploring gbtree, gblinear and dart, 0.3 as subsample, from 0.3 and 0.6, 3 as

logistic regression[16] as a feature-based baseline. These were trained using the 119 features extracted using the method described in Section 4.2. The evaluation of these models is shown in Table 5.

In order to characterize human level performance on the task, we report standard AES evaluation metrics for the human raters (column "raters" in Table 5), namely accuracy, adjacent accuracy, F1-score, and QWK.[17] Those metrics were calculated by a direct comparison between the ratings of one of the two evaluators and the reference CEFR levels for each essay. Those results show that the task of identifying the CEFR level of an essay is hard, even for humans. However, the adjacent accuracy of 0.99 clearly shows that the identification gap is typically up to one level. In the same direction, the QWK points out that once the ordinality existing between CEFR levels is considered, the agreement among raters is remarkably strong. As expected, none of our models achieved results competitive with human performance.[18]

Among the AES models explored, the transformer-based CamemBERT achieved the best values. Despite this performance, it can be seen that there is still room for improvement when comparing the results with the evaluation by experts (column raters). Considering that the transformers model performs in a range between raters and XGBoost, it is interesting to remark that the transformers model is closer to the raters' performance when we consider the ordinality relation between levels (i.e., QWK and $Accuracy_{Adjacent}$). Focusing on the ability of models to discriminate specific levels, the fine-tuned version of CamemBERT emerged as a model of better performance. Moreover, the logistic model is clearly a weak baseline. Interestingly, at the C2 level, which was the most challenging for all

three models, XGBoost suffers from a catastrophic failure, achieving even lower performance than the Logistic model. This general weak performance is not entirely surprising, as texts at this level tend to explore language idiosyncrasies, to be precise, to have a very coherent and organized structure, etc. In contrast, the beginner levels (i.e., A1 and A2), for which transformers and XGBoost models had close results, is characterized by texts with simple vocabulary and grammatical structures.[19]

As TCFLE-8 is a new corpus for the French language, we cannot fairly compare our results with previous works, due to the considerable difference in corpus size. In the French AES literature, we identified only two papers focusing on L2 proficiency identification. First, Parslow (2015a), who used a corpus of 200 essays to train a Naive Bayes classifier and reported F1-scores ranging from 0.51 to 0.74 for the levels A1 to B2. Second, Ranković et al. (2020) used CamemBERT intermediate layers as features to predict level in a corpus of 100 essays and reported MSE ranging from 0.35 to 0.55.

## 6   Final Remarks

In this work, we presented TCFLE-8, a corpus of 6,569 candidates' essays written during the French knowledge test (TCF), with 8 different languages of habitual use. This paper described the data gathering by France Education International (FEI), data cleaning, anonymization, and annotation performed to compile this corpus, which is the largest French corpus targeting French as a foreign language for AES. This corpus, along with its metadata (i.e., essays, metadata and annotation) is available to the community. We also described the learners' proficiency in the corpus using numerous linguistic variables related to SLA. This description confirms that these linguistic features could capture developmental patterns in TCFLE-8 in a similar fashion to other learner corpora.

Exploring TCFLE-8 for AES, we applied different machine learning algorithms. CamemBERT appears to be more accurate and XGBoost, a feature-based model, achieved similar results at beginner level. This raises a question about what features should explored for better describing the intermediate and advanced levels. Interestingly, part of this answer may come from the transformer model

---

max depth, from 3, 6 and 9, 0 as max delta step, exploring 0, 5 and 10, 1 as min child weight, from 1, 3 and 10, 0.1 as eta, exploring 0.01, 0.1, 0.3, 0.5 and 0.7, 1 as gamma, from 0, 1, 10, lossguide as grow policy, exploring lossguide and depthwise, multi softmax as objective function and 50 estimators.

[16]For the logistic regression, we explored the following hyperparameters: penalty from l2 or none, C from 1, 10, 100, max interaction from 100 or 300, and multi class process from multinomial or one-vs-rest. After this short exploration, we set the solver as lbfgs, l2 as the penalty, the C and the max interaction as 100 and 300, and the class processing as one-vs-rest.

[17]For a transparent presentation of our results, Appendix C shows the confusion matrices of the transformer-based model.

[18]Note that the scores in the "raters" column are inflated, because the rating assigned by each rater contributes to the final CEFR score of each essay.

[19]We compared the results of training the models on the anonymized corpus with the corresponding models trained on the original corpus (before anonymization) and no statistical difference was identified.

|  | **CamemBERT** | **XGBoost** | **Logistic** | **Raters** |
|---|---|---|---|---|
| QWK | **0.88** (0.01) | 0.79 (0.02) | 0.69 (0.02) | 0.93 (0.01) |
| Accuracy | **0.57** (0.01) | 0.46 (0.01) | 0.37 (0.01) | 0.76 (0.01) |
| Accuracy$_{\text{Adjacent}}$ | **0.98** (0.01) | 0.92 (0.02) | 0.80 (0.01) | 0.99 (0.01) |
| F1$_{\text{weighted}}$ | **0.56** (0.01) | 0.46 (0.02) | 0.36 (0.02) | 0.76 (0.01) |
| A1$_{\text{F1}}$ | **0.63** (0.01) | 0.59 (0.04) | 0.54 (0.06) | 0.76 (0.02) |
| A2$_{\text{F1}}$ | **0.57** (0.04) | 0.53 (0.01) | 0.40 (0.05) | 0.76 (0.03) |
| B1$_{\text{F1}}$ | **0.56** (0.04) | 0.45 (0.05) | 0.32 (0.02) | 0.75 (0.01) |
| B2$_{\text{F1}}$ | **0.56** (0.04) | 0.43 (0.03) | 0.34 (0.03) | 0.76 (0.02) |
| C1$_{\text{F1}}$ | **0.56** (0.04) | 0.42 (0.03) | 0.30 (0.05) | 0.77 (0.02) |
| C2$_{\text{F1}}$ | **0.48** (0.09) | 0.19 (0.07) | 0.31 (0.02) | 0.80 (0.04) |

Table 5: Average and standard deviation of the evaluation score using of using the TFCFL-8 for AES and the performance of human raters

itself (e.g. through a probing approach (Tenney et al., 2019)).

Finally, TCFLE-8 was portrayed in this paper as a corpus for French AES but its properties allow for different applications in NLP, SLA and educational studies. It may be a valuable corpus for pedagogical material development, whether they be dictionaries (Longman, 2002), activities focusing on common learner difficulties and errors (Kaszubski, 1998; Reppen, 2010), computer-assisted language learning software (Granger, 2003) or L2 writing aids (Link et al., 2014). Activities of data-driven learning in language class (Friginal, 2018) could also take advantage of this corpus. With 8 different languages of habitual use, this corpus could also be beneficial for cross-linguistic studies such as transfer mechanisms and L1 influence on L2 production (Golden et al., 2017; Werner et al., 2020), and for automatic native language identification (Tetreault et al., 2013). Another possible application for this new corpus is the one of errors detection and correction (Dahlmeier et al., 2013), that we are currently investigating as future work on TCFLE-8.

## 7 Limitations

Several normalization steps were applied in order to develop a coherent corpus for AES, aiming to compile a high quality corpus illustrating the proficiency levels with learner productions on which professional raters would agree. As a consequence, some potentially interesting cases were removed. This concerns for example texts on the borderline between two levels. Although they are interesting cases as they could support studies on understanding of the level boundaries, we opted for a corpus that represents the texts of each level. TCFLE-8

is a corpus designed for supporting the research in French as a foreign language, including AES. In this work, the focus is on the corpus compilation. Despite the initial tests performed here, our goal does not include an exhaustive verification of the corpus' applications nor an evaluation of various AES approaches. Finally, we do not intent nor recommend using TCFLE-8 for a fully-automated evaluation environment but to improve writing assessment in French as a foreign language.

## Acknowledgements

This research has been funded by the Fonds de la Recherche Scientifique de Belgique (F.R.S.-FNRS) under the grant MIS/PGY F.4518.21 and by a research convention with France Éducation International. Computational resources have been provided by the Consortium des Équipements de Calcul Intensif (CÉCI), funded by the Fonds de la Recherche Scientifique de Belgique (F.R.S.-FNRS) under Grant No. 2.5020.11 and by the Walloon Region

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

# A   Description of the learner and candidate corpora

Table 6 in this appendix provides a summary of candidate corpora collected from L2 certification exams and learner corpora targeting French for a comparison with TCFLE-8.

| CANDIDATE CORPUS FROM L2 CERTIFICATIONS | | | | | | |
|---|---|---|---|---|---|---|
| CANDIDATE CORPUS | L2 | NUMBER OF L1 | LEVEL | NUMBER OF TEXTS | TESTING INSTITUTION | ANNOTATION |
| Open Cambridge Learner Corpus (Open CLC) | ENG | 7 | all levels | 10,000 | Cambridge Assessment English | auto: POS |
| CLC - First Certificate English (CLC-FCE) | ENG | - | all levels | 1,238 | Cambridge Assessment English | auto: POS and syntactic manual: errors |
| ETS Corpus of Non-Native Written English | ENG | 11 | all levels | 12,100 | Education Testing Services (TOEFL) | raw |
| MERLIN | CZE GER ITA | - | A1-B2 A1-C1 A1-B2 | 2,290 | ÚJOP TELC | auto: POS manual: errors, syntactic, CEFR related |
| ASK | NNO | 10 | B1-B2 | 1,936 | Norwegian Language Test | auto: POS manual : errors, syntactic |
| COPLE2 | POR | 14 | A1-C1 | 966 | CAPLE ICLP | auto: POS manual: errors |
| **TCFLE-8** | FRE | 8 | all levels | 6,500 | France Education International | |
| LEARNER CORPORA TARGETING FRENCH | | | | | | |
| LEARNER CORPUS | L2 | L1 | LEVEL | NUMBER OF WORDS | COLLECTION CONTEXT | ANNOTATION |
| French Interlanguage Database (FRIDA) | FRE | ENG DUT others | int - adv | 450,000 | language class (univeristy) | manual: errors |
| Learner Corpus French | FRE | DUT | B2-C1 | 500,000 | language class (university) | - |
| Chy-FLE Hellas-FLE | FRE | GRE | int - adv | 150,000 | language class (university) L2 high-school exam | manual: grammatical constituents order |
| Corpus Interlangue (CIL) | FRE ENG | ARA Madarin ENG SPA FRE | B1-C1 | - (115 txt) | texts, read aloud and interviews from 115 students (university) | no annotation |
| Corpus Ecrit de Français Langue Etrangère | FRE | SWE | deb - adv | 100,000 | language class (high-school) | auto: POS manual: errors |
| Word Reference Corpus | FRE ENG SPA ITA | - | not evaluated | FFL: 4M. | forum posts of Word reference website | no annotation |
| Dire Autrement | FRE | ENG | int - adv | 50,000 | language class (university) | manual: lexical errors |
| **TCFLE-8** | FRE | ENG,ARA, SAP,RUS, POR,KAB, CHI,JPN | A1-C2 | 580,000 | written production of TCF certification (France Education International) | auto: text-level annotation |

Table 6: Candidate corpora and French learner corpora

# B   (Pseudo-)anonymization

For (pseudo-)anonymization, we start by applying the MAPA tool in the entire corpus. Next, we followed Volodina et al. (2019) by selecting 200 random texts and evaluating them manually to assess MAPA's output. In addition, we controlled the same amount of text from each level because different levels can affect the system differently. However, contrarily to Volodina et al. (2019), we also evaluate whether anonymization is appropriate. The reason for this stricter approach was to measuring the of distorting caused by the (pseudo-)anonymization step.

The MAPA's evaluation was carried out by two independent French native speaker. Each one evaluated 100 essays. Later, a third evaluator double-checked the 200 essays searching for inconsistencies, which were fixed after discussion with the other evaluators. During this assessment, we identify standard errors. These errors, described in Section 3.3, were automatically corrected after we identified their patterns of occurrence.

The results of this evaluation is shown in Table 7. The first observation is about the ability to fully identify an entity where MAPA presents difficulty. However, we point out that partial anonymization is already considered correct. In addition, a small number of errors are caused by the entity type, as exemplified by the close scores in the partial matching columns in the table, where the only distinction is the consideration of the entity and span or just the span. In this evaluation, we highlight two scores: accuracy and F2. The first takes into account the words that have been correctly identified as non-anonymized. The second, on the other hand, is a variation of the F-score where recall receives a greater weight. F2 represents the interest of coverage but without a significant loss in precision. Given the difference between these scores, and the recall and precision values, we notice that MAP tends to overdo, identifying more terms than needed for anonymization. Although the anonymization method generates an abundance of edits in the text, it ensures quality in the process and in the protection of the privacy of the writers.

| | span-based | | entity-based |
|---|---|---|---|
| | **Partial match** | **Exact match** | **Partial match** |
| *Accuracy* | 0.97 | 0.96 | 0.97 |
| *Precision* | 0.55 | 0.47 | 0.50 |
| *Recall* | 0.86 | 0.68 | 0.85 |
| *F1* | 0.67 | 0.56 | 0.63 |
| *F2* | 0.77 | 0.63 | 0.75 |

Table 7: Result of the anonymization evaluation

## C  Confusion matrices of transformer-based AES model

Table 8 shows the results for each of the 5 repetitions (see Section 5) of the AES model based on transformers (column CamemBert in Table 5).

**Prediction**

| Gold | A1 | A2 | B1 | B2 | C1 | C2 |
|---|---|---|---|---|---|---|
| A1 | 48 | 26 | 1 | 0 | 0 | 0 |
| A2 | 10 | 57 | 65 | 2 | 0 | 0 |
| B1 | 0 | 11 | 100 | 37 | 1 | 0 |
| B2 | 0 | 0 | 21 | 99 | 22 | 1 |
| C1 | 0 | 0 | 1 | 44 | 53 | 11 |
| C2 | 0 | 0 | 0 | 1 | 25 | 21 |

(a) Prediction of Run 1

**Prediction**

| Gold | A1 | A2 | B1 | B2 | C1 | C2 |
|---|---|---|---|---|---|---|
| A1 | 51 | 21 | 0 | 0 | 0 | 0 |
| A2 | 30 | 87 | 19 | 0 | 0 | 0 |
| B1 | 1 | 29 | 76 | 38 | 2 | 0 |
| B2 | 0 | 0 | 24 | 63 | 55 | 0 |
| C1 | 0 | 0 | 1 | 13 | 87 | 9 |
| C2 | 0 | 0 | 0 | 1 | 31 | 19 |

(b) Prediction of Run 2

**Prediction**

| Gold | A1 | A2 | B1 | B2 | C1 | C2 |
|---|---|---|---|---|---|---|
| A1 | 37 | 30 | 5 | 0 | 0 | 0 |
| A2 | 8 | 68 | 54 | 4 | 0 | 0 |
| B1 | 1 | 21 | 84 | 40 | 2 | 0 |
| B2 | 0 | 1 | 14 | 80 | 42 | 7 |
| C1 | 0 | 0 | 2 | 17 | 62 | 30 |
| C2 | 0 | 0 | 0 | 2 | 17 | 29 |

(c) Prediction of Run 3

**Prediction**

| Gold | A1 | A2 | B1 | B2 | C1 | C2 |
|---|---|---|---|---|---|---|
| A1 | 20 | 49 | 0 | 0 | 0 | 0 |
| A2 | 2 | 93 | 41 | 2 | 0 | 0 |
| B1 | 1 | 28 | 99 | 17 | 2 | 0 |
| B2 | 0 | 0 | 40 | 74 | 31 | 0 |
| C1 | 0 | 0 | 3 | 30 | 71 | 10 |
| C2 | 0 | 0 | 1 | 1 | 32 | 10 |

(d) Prediction of Run 4

**Prediction**

| Gold | A1 | A2 | B1 | B2 | C1 | C2 |
|---|---|---|---|---|---|---|
| A1 | 47 | 16 | 3 | 0 | 0 | 0 |
| A2 | 25 | 84 | 27 | 1 | 0 | 0 |
| B1 | 0 | 44 | 66 | 32 | 3 | 1 |
| B2 | 0 | 2 | 21 | 85 | 35 | 2 |
| C1 | 0 | 0 | 0 | 29 | 70 | 18 |
| C2 | 0 | 0 | 0 | 0 | 18 | 28 |

(e) Prediction of Run 5

Table 8: Predictions of the 5 repetions of the CamemBert models fine-tuned to TCFLE-8

## C.1 Correlated features

In this section, we list the features identified by the feature selection method presented in Section 4.2. Table 9 list as features and their correlations, while Figure 2 plots the distribution of some feature across the six CEFR levels.

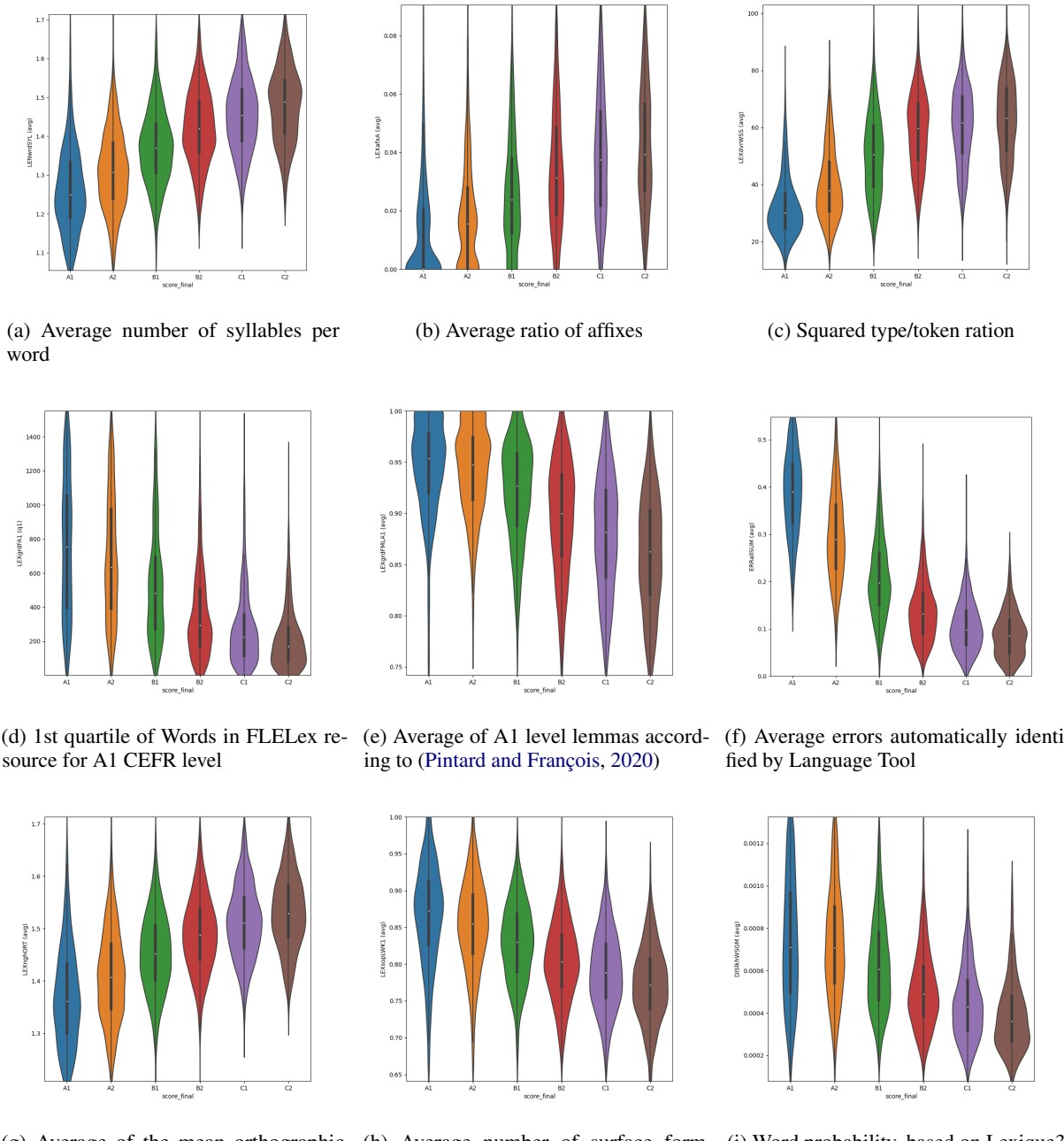

(a) Average number of syllables per word

(b) Average ratio of affixes

(c) Squared type/token ration

(d) 1st quartile of Words in FLELex resource for A1 CEFR level

(e) Average of A1 level lemmas according to (Pintard and François, 2020)

(f) Average errors automatically identified by Language Tool

(g) Average of the mean orthographic Levenstein distance (based on Lexique3)

(h) Average number of surface form words in the top 1000 words of Lexique3

(i) Word probability, based on Lexique3

Figure 2: Violin plot of some of the top correlated features through the 6 CEFR levels

| Variable Family | List of features and their correlations with essay CEFR level |
|---|---|
| Graded features | $LEXgrdFMLA1_{std}$ (0,463), $LEXgrdFSOOUA1_{10P}$ (-0,41), $LEXgrdFA2_{min}$ (-0,416), $LEXgrdFA1_{10P}$ (-0,529), $LEXgrdFSOOUC1_{var}$ (0,402), $LEXgrdFA2_{20P}$ (-0,532), $LEXgrdBA1_{kurtosis}$ (-0,418), $LEXgrdBA1_{skewness}$ (0,417), $LEXgrdFSOOUA1_{std}$ (0,457), $LEXgrdFSOOUC1_{std}$ (0,402), $LEXgrdFMLA1_{var}$ (0,463), $LEXgrdFA1_{20P}$ (-0,56), $LEXgrdFA2_{q1}$ (-0,531), $LEXgrdFB2_{10P}$ (-0,441), $LEXgrdFC1_{20P}$ (-0,403), $LEXgrdFA2_{10P}$ (-0,497), $LEXgrdFB1_{10P}$ (-0,45), $LEXgrdFSOOUA1_{avg}$ (-0,464), $LEXgrdFSOOUA1_{var}$ (0,457), $LEXgrdFB2_{20P}$ (-0,483), $LEXgrdFB1_{20P}$ (-0,497), $LEXgrdFB2_{q1}$ (-0,485), $LEXgrdFMLA1_{10P}$ (-0,411) |
| Lexical diversity | $LEXdvrWLR_{avg}$ (0,545), $LEXdvrFSS_{avg}$ (0,559), $LEXdvrWSS_{avg}$ (0,587), $LEXdvrFSR_{avg}$ (0,559), $LEXdvrVLRW_{avg}$ (0,433), $LEXdvrWLC_{avg}$ (0,545), $LEXdvrNSR_{avg}$ (0,54), $LEXdvrNSC_{avg}$ (0,54), $LEXdvrVLR_{avg}$ (0,478), $LEXdvrVSU_{avg}$ (0,451), $LEXdvrVLS_{avg}$ (0,478), $LEXdvrWSR_{avg}$ (0,587), $LEXdvrVLSW_{avg}$ (0,433), $LEXdvrVLCW_{avg}$ (0,433), $LEXdvrVLC_{avg}$ (0,478), $LEXdvrWLS_{avg}$ (0,545), $LEXdvrVSS_{avg}$ (0,482), $LEXdvrFLR_{avg}$ (0,558), $LEXdvrFLC_{avg}$ (0,558), $LEXdvrVSRW_{avg}$ (0,449), $LEXdvrFSC_{avg}$ (0,559), $LEXdvrFLS_{avg}$ (0,558), $LEXdvrNLC_{avg}$ (0,544), $LEXdvrNLS_{avg}$ (0,544), $LEXdvrVSSW_{avg}$ (0,449), $LEXdvrNSS_{avg}$ (0,54), $LEXdvrVSR_{avg}$ (0,482) |
| Lexical errors | $ERRallSUM_{avg}$ (-0,771) |
| Lexical Frequency | $LEXfrqLNL_{20P}$ (0,446), $LEXfrqLNS_{q1}$ (0,46), $LEXfrqFCL_{20P}$ (0,448), $LEXfrqFNL_{20P}$ (0,41), $LEXfrqLCS_{q1}$ (0,496), $LEXfrqFCL_{q1}$ (0,464), $LEXfrqLCL_{20P}$ (0,488) |
| Lexical sophistication | $LEXsopFK1_{var}$ (0,419), $LEXsopLWK1_{avg}$ (-0,443), $LEXsopLWK1_{skewness}$ (0,417), $LEXsopLWK1_{var}$ (0,427), $LEXsopLWK1_{std}$ (0,427), $LEXsopFK1_{avg}$ (-0,429), $LEXsopFK1_{std}$ (0,419) |
| LexMorphology features | $LEXafxA_{avg}$ (0,442) |
| Orthographic neighbors | $LEXnghNUM_{10P}$ (-0,448), $LEXnghPHO_{iqr}$ (0,426), $LEXnghPHO_{90P}$ (0,417), $LEXnghORT_{median}$ (0,407), $LEXnghNUM_{20P}$ (-0,526), $LEXnghNUMF_{20P}$ (-0,436), $LEXnghFRQ_{q1}$ (-0,431), $LEXnghPHO_{std}$ (0,432), $LEXnghPHO_{q3}$ (0,425), $LEXnghORT_{iqr}$ (0,504), $LEXnghAVGF_{20P}$ (-0,42), $LEXnghORT_{80P}$ (0,526), $LEXnghORT_{90P}$ (0,492), $LEXnghORT_{avg}$ (0,519), $LEXnghNUMF_{10P}$ (-0,436), $LEXnghPHO_{avg}$ (0,481), $LEXnghNUM_{q1}$ (-0,515), $LEXnghPHO_{max}$ (0,472) |
| Word length | $LENwrdSYL_{80P}$ (0,416), $LENwrdSYL_{q3}$ (0,449), $LENwrdLETTERS_{var}$ (0,415), $LENwrdSYL_{max}$ (0,468), $LENwrdLETTERS_{rsd}$ (0,406), $LENwrdSYL_{dolch}$ (0,414), $LENwrdSYL_{std}$ (0,517), $LENwrdSYL_{avg}$ (0,558) |
| Tense features | $SYNtnsfINDP_{avg}$ (-0,422) |
| Text likelihood | $DISlkhVSM_{kurtosis}$ (0,452), $DISlkhVSML_{avg}$ (-0,411), $DISlkhWLM_{q1}$ (-0,438), $DISlkhFSM_{20P}$ (-0,41), $DISlkhVLM_{min}$ (-0,424), $DISlkhWLGM_{avg}$ (-0,469), $DISlkhFSM_{median}$ (-0,401), $DISlkhNSM_{kurtosis}$ (0,401), $DISlkhWSM_{20P}$ (-0,48), $DISlkhVLGM_{avg}$ (-0,4), $DISlkhFSM_{skewness}$ (0,409), $DISlkhVLM_{20P}$ (-0,451), $DISlkhWSML_{avg}$ (-0,509), $DISlkhWLM_{10P}$ (-0,416), $DISlkhFLM_{skewness}$ (0,455), $DISlkhVSM_{20P}$ (-0,453), $DISlkhALM_{10P}$ (-0,424), $DISlkhASM_{20P}$ (-0,422), $DISlkhFLM_{rsd}$ (0,438), $DISlkhVSM_{median}$ (-0,417), $DISlkhASM_{min}$ (-0,406), $DISlkhWLM_{median}$ (-0,419), $DISlkhWLM_{rsd}$ (0,4), $DISlkhVSM_{min}$ (-0,424), $DISlkhALM_{20P}$ (-0,402), $DISlkhWSM_{median}$ (-0,455) |

Table 9: Correlation between features and CEFR level grouped by linguistic variable family. For the name of the features, see https://cental.uclouvain.be/fabra/docs.html.