# OpenReview forum: "TCFLE-8: a Corpus of Learner Written Productions for French as a Foreign Language and its Application to Automated Essay Scoring"
_EMNLP/2023/Conference — EMNLP 2023 Main_

### Official Review · Reviewer_cs99 · 2023-07-29

**Soundness:** 5

**Excitement:**

4: Strong: This paper deepens the understanding of some phenomenon or lowers the barriers to an existing research direction.

**Paper Topic And Main Contributions:**

This paper describes a new resource for French-language AES research to be released publicly.  The corpus consists of essays across a range of CEFR levels, from the Test de Connaissance de Francais certification exam.  The paper describes distributional statistics of the corpus, criteria for inclusion of essays, available metadata, and linguistic features made available as part of the resource.  The paper also presents baseline AES results using deep learning and feature-based machine learning

**Reasons To Accept:**

- The corpus as described is sizeable and of high quality.  The rigorous scoring protocol and useful metadata will make this an important resource for researchers working on AES, especially those focused on the French language
- The baseline scoring results described here will be a helpful starting point for researchers wishing to apply novel methods to this task
- The literature review is comprehensive and on-point
- The paper is well-crafted and clear in its presentation and organization

**Reasons To Reject:**

I have no significant concerns about the paper that would lead me to recommend rejection.  However

- I was surprised by the variability in performance between methods in Table 2.  The high accuracy for Camembert, coupled with much lower QWK, suggests that it may have a serious majority-class bias.  Statistics on the predicted distribution would be useful information.
- I concur with the authors' note in section 7 (Limitations) that the criteria for exclusion of essays that are not clear examples of one score/level or another represent a significant gap to consider



**Reproducibility:**

4: Could mostly reproduce the results, but there may be some variation because of sample variance or minor variations in their interpretation of the protocol or method.

**Reviewer Confidence:**

4: Quite sure. I tried to check the important points carefully. It's unlikely, though conceivable, that I missed something that should affect my ratings.

---

> ### Author Rebuttal · Authors · 2023-08-29
>
> We greatly appreciate your insightful review of our paper. Your feedback is invaluable in enhancing the quality and impact of our work, and we are pleased to see that the overall assessment is positive. Your acknowledgment of the potential impact on AES research, particularly in the context of the French language, motivates us to continue our efforts in facilitating advancements in this field.
>
> We appreciate your observation regarding the variability in performance between methods in Table 2. Your concern about potential majority-class bias in the Camembert model is valid, and we apologize for not including the predicted distribution statistics in the current version of the paper. We will promptly address this by providing the statistics on the predicted distribution (i.e., confusion matrix) to enhance the transparency and understanding of the model's performance. Additionally, we highlight that we tried to make a stratified sample as much as possible. However, the efforts to stratify the corpus ended up interfering with the final size of the corpus. We therefore opted for a larger corpus with an imbalance in the rare classes in the corpus.
>
> Your agreement with our note in Section 7 about the criteria for the exclusion of essays is noted. We acknowledge that this represents a gap, and your feedback reinforces the importance of this aspect that may be explored in the future. We will further elaborate on the considerations and implications of this exclusion criterion in the revised version of the paper.
>
> We're deeply grateful for your thoughtful review and constructive feedback. Your observations will enhance the clarity and comprehensiveness of our paper. Thank you once again for your time and consideration.

---

### Official Review · Reviewer_5vDV · 2023-08-01

**Soundness:** 5

**Excitement:**

4: Strong: This paper deepens the understanding of some phenomenon or lowers the barriers to an existing research direction.

**Paper Topic And Main Contributions:**

This work presents a new corpus for automated essay scoring for French.
The corpus contains the followings:
- 6,569 essays
- CEFR ratings (A1-C2)
- L1 labels
- gender labels
- CEFR level assigned to examinees
- many annotations automatically assigned

It provides the way of creating the corpus including explanation of the test, cleaning, and distribution of L1.

The paper also provides the base line results using CamemBERT, XGBoost, and logistic regression.
The correlation of annotated variables and CEFR levels is also reported.

Last but not least, the related work section a comprehensive review on AES corpora including those for English and French.

**Questions For The Authors:**

0. What does "LE" of TCFLE-8 stand for? Does "8" mean eight L1 languages?


**Reasons To Accept:**

- According to Table 3, the proposed corpus is superior to previous corpora, which are not suitable for AES because of the rating span.
- Rich metadata and annotations.
- Comprehensive related work.
- The baselines are provided.

**Reasons To Reject:**

I did not find any flaw that prevents the paper from being published.

**Reproducibility:**

4: Could mostly reproduce the results, but there may be some variation because of sample variance or minor variations in their interpretation of the protocol or method.

**Reviewer Confidence:**

3: Pretty sure, but there's a chance I missed something. Although I have a good feel for this area in general, I did not carefully check the paper's details, e.g., the math, experimental design, or novelty.

**Typos Grammar Style And Presentation Improvements:**

- l.319 Masumi Ono et al -> Ono et al (Masumi is her first name)

---

> ### Author Rebuttal · Authors · 2023-08-29
>
> We are pleased to note your recognition of the main contributions of our paper. Furthermore, we are encouraged by your positive evaluation of our work's strengths.
>
> Regarding your question about "LE" in "TCFLE-8," we appreciate your inquiry and will precise it in the final version. TCF stands for “Test de connaissance du français” and FLE stands for “français langue étrangère” (french as a foreign language) and "8" indeed refers to the eight different L1 languages included in the corpus.
>
> We thank you for your attention to detail in identifying the correct naming convention for the authors. We will make the necessary correction to accurately cite the reference as "Ono et al" rather than "Masumi Ono et al."
>
> We sincerely appreciate your thorough evaluation and positive feedback. Your insights have contributed to improving the quality and impact of our paper. Thank you for your time and consideration.

---

### Official Review · Reviewer_p5TN · 2023-08-02

**Soundness:** 5

**Excitement:**

4: Strong: This paper deepens the understanding of some phenomenon or lowers the barriers to an existing research direction.

**Missing References:**

N/A

**Paper Topic And Main Contributions:**

The paper contributes to the improvement automated essay scoring for French and introduces a new valuable corpus consisting of 6.5k essays collected in a well-known certification exam for French (the official French Knowledged Test, TCF) that are related to the CEFR levels. The paper also discusses how certain key linguistic properties of the essays relate to the learners’ proficiency as reflected in the corpus data.

**Questions For The Authors:**


A) In lines 527-529 the manuscript states "For example, measures of word length have been known to be good predictors of proficiency levels", followed by two references (Ferris, 1994 and Grant and Gither, 2000). These reviewer is not familiar with these cited references, unfortunately, but I have the following three questions for clarity:

A) The two references appear rather dated, are there any other more recent studies backing up this statement?

B) Do these two references refer to French specifically, to the exclusion of other languages (e.g. English), so that the statement is applicable and relevant here?

C) Do these two references apply to learner language (i.e. French, as noted in the previous question b), and written essays in certification examinations in particular (regardless of the CEFR level)? This seems to be a very specific setting in which this statement must apply.




D) It is unclear to me what the author(s) mean(s) in line 349 when they mention "atypical candidates": this should be explained and potentially exemplified, although I wonder whether it is a badly chosen phrase, that may carry some negative overtones if not outright stigma with it. In particular, I would argue that being "on the borderline between levels" is not (necessarily or always) the same as being an "atypical candidate", i.e. there's no identity and possibly also no correlation between these two concepts. The author(s) would do well to consider a suitable (and clearer, more intuitive) alternative term here that does not raise controversy and does not appear to stigmatize candidates, whatever their circumstances and results.



E) Statement in lines 542-544: "To conclude, if this analysis confirms existing research findings, it also points out that TCFLE-8 may be helpful for SLA studies.": Yes, of course, I would agree in principle with the claim, but how can this be done, exploring which issues specifically?? Otherwise like this the statement is rather empty.




**Reasons To Accept:**

An interesting and well-written paper that addresses a relevant issue with originality and a sound and valid methodology, providing clear and useful results. In addition, the paper describes in detail a novel and very valuable corpus of approximately 6.5k language certification essays in French that has been carefully curated and is accompanied by interesting metadata.

**Reasons To Reject:**

None in particular, or that struck me as definitely negative upon reading the manuscript.

**Reproducibility:**

5: Could easily reproduce the results.

**Reviewer Confidence:**

4: Quite sure. I tried to check the important points carefully. It's unlikely, though conceivable, that I missed something that should affect my ratings.

**Typos Grammar Style And Presentation Improvements:**


Minor comments and suggestions that the authors might consider to improve the manuscript upon revision:


- The paper uses the term "usual language" a number of times, which is duly explained in footnote 5 as follows: "The usual language is the language the candidate indicates as the one they usually use.". As far as I know, the standard (and probably preferred, also on the grounds of being more transparent and intuitive) term to indicate this same concept is "language of habitual use" (of the candidates / students / examinees). It is understood that this is not (necessarily / always) the same as first / native language, which is fine.

- The manuscript explains the meaning and full form of CEFR in line 146, but the acronym is already used in the abstract (line 018) and in line 093 before then: acronyms/abbreviations should be explained on first occurrence with their full form, and then be used consistently thereafter.

- Footnote 2: I appreciate that the title of the referenced learner corpus collection is clickable and has a built-in link to the online resource, but probably the footnote should rather (or also?) give the full URL.

- Typo in line 138: "Corpora built for AES can focus _of_ specific dimensions" >>> "Corpora built for AES can focus _ON_ specific dimensions"

- Section 2.1.1 on candidate corpora focuses on English, so what is the relevance of this to the study presented in the paper, which focuses on French? Probably the same information can be given in a more compact from in a table, if it is deemed relevant?

- Similarly, Section 2.1.2 on corpora from language classes focuses again on English; while it's clear that this is the dominant language, the direct relevance of this information for the paper that is devoted to French is largely unclear. As suggested above for similar reasons for section 2.1.1, probably the same information (about English, as background for comparison purposes) can be presented more succinctly in table format.

- Lines 168-169 mention "three recent candidate corpora", but they go as far back as 2013 (at least), so the use of the adjective "recent" does not seem (equally) justified for all of them.

- Typo in line 202: "Appendix A provides _a_ additional detailed description" >>> "Appendix A provides _AN_ additional detailed description"

- Typo in line 342: "which _humans_ raters" >>> "which _HUMAN_ raters"

- Line 346 mentions that an "empirical evaluation" was peformed, but it remains unclear how this was conducted; it would be useful and interesting for the readers to give at least a brief description of this evaluation and what it main findings were that justified removing all essays with a standard residual value greated than 4. In particular, how many essays from the whole initial collections were dropped, as a result of this decision?

- It is unclear to me what the author(s) mean(s) in line 349 when they mention "atypical candidates": this should be explained and potentially exemplified, although I wonder whether it is a badly chosen phrase, that may carry some negative overtones if not outright stigma with it. In particular, I would argue that being "on the borderline between levels" is not (necessarily or always) the same as being an "atypical candidate", i.e. there's no identity and possibly also no correlation between these two concepts. The author(s) would do well to consider a suitable (and clearer, more intuitive) alternative term here that does not raise controversy and does not appear to stigmatize candidates, whatever their circumstances and results.

- Potential typo in lines 560-561, to be checked: "Any essay that does not fit XXX of these two criteria..." >>> "Any essay that does not fit _EITHER_ of these two criteria..."???

- There's a clear and serious inconsistency (that can be easily resolved) between line 368 (which states "As the top six [languages] were all European ones,,,"), which does not correspond to the statement in line 454, which also mentions Arabic, Russian and Kabyle (in addition to Chinese and Japanese), as follows: "As for the usual language, the corpus covers 8 languages, as described in Section 3.2: English, Arabic, Spanish, Russian, Portuguese, Kabyle, Chinese, and Japanese...". These explanations of the languages covered by the corpus in terms of "usual languages" (or, perhaps preferably, "languages of habitual use", as noted above) of the learners/examinees seem to be inconsistent and in conflict. Also make sure that there are no other incompatible or confusing statements in this regard elsewhere in the manuscript.

- In lines 527-529 the manuscript states "For example, measures of word length have been known to be good predictors of proficiency levels", followed by two references (Ferris, 1994 and Grant and Gither, 2000). These reviewer is not familiar with these cited references, unfortunately, but I have the following three questions for clarity:

a) The two references appear rather dated, are there any other more recent studies backing up this statement?

b) Do these two references refer to French specifically, to the exclusion of other languages (e.g. English), so that the statement is applicable and relevant here?

c) Do these two references apply to learner language (i.e. French, as noted in the previous question b), and written essays in certification examinations in particular (regardless of the CEFR level)? This seems to be a very specific setting in which this statement must apply.


- Line 530 mentions the acronym "TTR", without explanation. I assume this is "type/token ratio", and in any case, the full form of the term/concept should be given, to avoid ambiguity and potential confusion on the part of the readers, especially if the acronym/concept is not used elsewhere.

- Statement in lines 542-544: "To conclude, if this analysis confirms existing research findings, it also points out that TCFLE-8 may be helpful for SLA studies.": Yes, of course, I would agree in principle with the claim, but how can this be done, exploring which issues specifically?? Otherwise like this the statement is rather empty.

- Typo in line 582: "between the ratings of one of the two _evaluator_ and the reference CEFR levels" >>> "between the ratings of one of the two _EVALUATORS_ and the reference CEFR levels"

- Typo in footnote 14: "... because the rating assigned by each rater _contribute_ to the final CEFR score..." >>> "... because the rating assigned by each rater _CONTRIBUTES_ to the final CEFR score..."

- Line 634: "anonymization" >>> "(pseudo-)anonymization"

- Possible typo in line 657: "Given the weak complementary observed between both solutions" >>> should "complementary" be "complementarity" instead?

- Typo in line 687: "However, this _lead_ to the removal" >>> "However, this _LED_ to the removal"

- Generally, Appendix B on (pseudo-)anonymisation is not well written, there are several obvious typos: that whole part should be carefully proofread to resolve the problems.

- Typo in line 1277: "Finally, Table 7 picture the amount of essays"??? This seems incorrect: proofread careafully and amend as required.

- Other typos are found in lines 1284-1285: proofread carefully and amend as required.

- Table 5: the label in the left-hand column should probably be "(CEFR) Level", rather than "Language" for clarity.

- Again Table 5: I would add to the caption (the rest stays as is): ", per (CEFR) level"

- Typo in Table 17: "Arab" >>> "Arabic" (for the language)

---

> ### Author Rebuttal · Authors · 2023-08-29
>
> We appreciate your thorough and insightful review of our paper. Your feedback and meticulous attention to detail are valuable in refining our work, and we are grateful for the positive assessment you've provided. We would like to address the points you raised:
>
> Your recommendations for using "language of habitual use" instead of "usual language," resolving acronym explanations, and addressing the inconsistencies in the language coverage descriptions are duly noted. Additionally, we will implement your suggestions to enhance the manuscript's readability and coherence.
>
> > A) In lines 527-529 the manuscript states "For example, measures of word length have been known to be good predictors of proficiency levels", followed by two references (Ferris, 1994 and Grant and Gither, 2000). These reviewer is not familiar with these cited references, unfortunately, but I have the following three questions for clarity:
> > A) The two references appear rather dated, are there any other more recent studies backing up this statement?
> > B) Do these two references refer to French specifically, to the exclusion of other languages (e.g. English), so that the statement is applicable and relevant here?
> > C) Do these two references apply to learner language (i.e. French, as noted in the previous  question b), and written essays in certification examinations in particular (regardless of the CEFR level)? This seems to be a very specific setting in which this statement must apply.
>
> We have selected these references to show that this is already known in the domain. Ferris (1994) and Grant and Gither (2000) address AES for English. And, for the sake of completeness, we’ll include the following references, addressing AES for Swedish, Japanese, and French languages, respectively.
> - Pilán, I., & Volodina, E. (2018, August). Investigating the importance of linguistic complexity features across different datasets related to language learning. In Proceedings of the Workshop on Linguistic Complexity and Natural Language Processing (pp. 49-58).
> - Hirao, R., Arai, M., Shimanaka, H., Katsumata, S., & Komachi, M. (2020, May). Automated essay scoring system for nonnative japanese learners. In Proceedings of the Twelfth Language Resources and Evaluation Conference (pp. 1250-1257).
> - Parslow, N. L. (2015). Automated Analysis of L2 French Writing: a preliminary study (Doctoral dissertation, Master’s thesis). University of Paris Diderot. doi: 10.13140/RG. 2.1. 2833.5204).
>
> Ferris (1994) applies to certification examinations in particular (i.e., TSOL), while the other references address learner corpora.
>
> > D) It is unclear to me what the author(s) mean(s) in line 349 when they mention "atypical candidates": this should be explained and potentially exemplified, although I wonder whether it is a badly chosen phrase, that may carry some negative overtones if not outright stigma with it. In particular, I would argue that being "on the borderline between levels" is not (necessarily or always) the same as being an "atypical candidate", i.e. there's no identity and possibly also no correlation between these two concepts. The author(s) would do well to consider a suitable (and clearer, more intuitive) alternative term here that does not raise controversy and does not appear to stigmatize candidates, whatever their circumstances and results.
>
> Thank you for the suggestion. We'll change it for candidates on the borderline between levels.
>
>
> > E) Statement in lines 542-544: "To conclude, if this analysis confirms existing research findings, it also points out that TCFLE-8 may be helpful for SLA studies.": Yes, of course, I would agree in principle with the claim, but how can this be done, exploring which issues specifically?? Otherwise like this the statement is rather empty.
>
> We'll precise it by replacing it with “In addition, while this analysis confirms existing research findings in AES, it also points out that TCFLE-8 may be helpful for new SLA studies.”
>
>
> > The paper uses the term "usual language" a number of times, which is duly explained in footnote 5 as follows: "The usual language is the language the candidate indicates as the one they usually use.". As far as I know, the standard (and probably preferred, also on the grounds of being more transparent and intuitive) term to indicate this same concept is "language of habitual use" (of the candidates / students / examinees). It is understood that this is not (necessarily / always) the same as first / native language, which is fine
>
> We selected this term because it reflects how the term used by FEI is typically translated to English in the scientific literature. However, we do agree that language of habitual use makes a clear statement. We'll update it.
>
>
> > The manuscript explains the meaning and full form of CEFR in line 146, but the acronym is already used in the abstract (line 018) and in line 093 before then: acronyms/abbreviations should be explained on first occurrence with their full form, and then be used consistently thereafter.
>
> Thanks for pointing it out. We'll correct this mistake.
>
>
> > Footnote 2: I appreciate that the title of the referenced learner corpus collection is clickable and has a built-in link to the online resource, but probably the footnote should rather (or also?) give the full URL.
>
> We add it as a built-in link due to space constraints as it's a long URL, but we'll include the entire URL for accessibility of people with software issues or using a paper version.
>
>
> > Section 2.1.1 on candidate corpora focuses on English, so what is the relevance of this to the study presented in the paper, which focuses on French? Probably the same information can be given in a more compact from in a table, if it is deemed relevant?
>
> Section 2.1.1 presents several well-known candidate corpora to situate TCFLE-8 among similar corpora. There are 2 English corpora, one targeting German, Italian and Czech, one Norwegian and one Portuguese. We’ll add a reference to Table 3 in the Appendix for an overlook of learner corpora comparable to TCFLE-8.
>
>
> > Similarly, Section 2.1.2 on corpora from language classes focuses again on English; while it's clear that this is the dominant language, the direct relevance of this information for the paper that is devoted to French is largely unclear. As suggested above for similar reasons for section 2.1.1, probably the same information (about English, as background for comparison purposes) can be presented more succinctly in table format.
>
> Section 2.1.2 presents learner corpora that have been used for AES, with one targeting the English language, the biggest learner corpus used for AES, and three others targeting Spanish, Swedish and Japanese. We’ll also add a reference to Table 3
>
>
> > Lines 168-169 mention "three recent candidate corpora", but they go as far back as 2013 (at least), so the use of the adjective "recent" does not seem (equally) justified for all of them.
>
> We'll remove the "recent" as we agree it is needless here.
>
>
> > Line 346 mentions that an "empirical evaluation" was peformed, but it remains unclear how this was conducted; it would be useful and interesting for the readers to give at least a brief description of this evaluation and what it main findings were that justified removing all essays with a standard residual value greated than 4. In particular, how many essays from the whole initial collections were dropped, as a result of this decision?
>
> We’ll provide a brief description of the final version.
>
>
> > Line 530 mentions the acronym "TTR", without explanation. I assume this is "type/token ratio", and in any case, the full form of the term/concept should be given, to avoid ambiguity and potential confusion on the part of the readers, especially if the acronym/concept is not used elsewhere.
>
> Yes, it's type/token ratio. We'll precise it.
>
>
> > Table 5: the label in the left-hand column should probably be "(CEFR) Level", rather than "Language" for clarity. Again Table 5: I would add to the caption (the rest stays as is): ", per (CEFR) level"
>
> We’ll fix the column head and the caption.
>
> Your careful evaluation adds credibility to our work and reinforces our commitment to delivering robust research outcomes. Your thorough review and insightful comments have been instrumental in guiding our revisions. We are dedicated to addressing your concerns and making the necessary improvements to enhance the overall quality of the paper. Thank you for your time and consideration.

---

### Meta-Review · Area_Chair_ABEQ · 2023-09-07

**Recommendation:** 5

**Metareview:**

This paper presents an original and valuable corpus for automatic essay scoring for French. The content and presentation of this paper is excellent, as reflected by three Soundness ratings of 5 and three Excitement ratings of 4. (Two reviewers rated its Reproducibility a 4 and one rated it a 5.) Other aspects of note are the paper's clarity, its comprehensive literature review, the quality of the corpus presented, as well as the inclusion of baseline results that set the stage for future work. None of the reviewers cited reasons to reject the paper, though all raised minor issues (e.g. grammatical errors, typos, unclear terminology) that were resolved during the author rebuttal period.

In light of this, only minor revisions, addressing reviewers’ comments and questions, need to be made to ensure this paper is camera ready.

---

### Decision · Program_Chairs · 2023-10-07

**Decision:**

Accept-Main

**Comment:**

This paper presents an original and valuable corpus for automatic essay scoring for French. The content and presentation of this paper is excellent, as reflected by three Soundness ratings of 5 and three Excitement ratings of 4. (Two reviewers rated its Reproducibility a 4 and one rated it a 5.) Other aspects of note are the paper's clarity, its comprehensive literature review, the quality of the corpus presented, as well as the inclusion of baseline results that set the stage for future work. None of the reviewers cited reasons to reject the paper, though all raised minor issues (e.g. grammatical errors, typos, unclear terminology) that were resolved during the author rebuttal period.

In light of this, only minor revisions, addressing reviewers’ comments and questions, need to be made to ensure this paper is camera ready.